# Predicting Falls in Musculoskeletal Rehabilitation: A Retrospective Multisite Study

**DOI:** 10.3390/healthcare11202805

**Published:** 2023-10-23

**Authors:** Luca Scarabel, Federica Scarpina, Graziano Ruggieri, Nicola Schiavone, Costanzo Limoni

**Affiliations:** 1Clinica Hildebrand, Centro di Riabilitazione, via Crodolo 18, 6614 Brissago, Switzerland; g.ruggieri@clinica-hildebrand.ch; 2Cliniche di Riabilitazione Ente Ospedaliero Cantonale (CREOC), 6986 Novaggio, Switzerland; nicola.schiavone@eoc.ch; 3“Rita Levi Montalcini” Department of Neurosciences, University of Turin, 10126 Turin, Italy; federica.scarpina@unito.it; 4I.R.C.C.S. Istituto Auxologico Italiano, U.O. di Neurologia e Neuroriabilitazione, Ospedale San Giuseppe, 28824 Piancavallo, Italy; 5Department of Economics, Health and Social Sciences, University of Applied Sciences and Arts of Southern Switzerland, 6928 Manno, Switzerland; costanzo.limoni@sunrise.ch

**Keywords:** fall risk, musculoskeletal rehabilitation, Functional Independence Measure

## Abstract

Background. Fall represents one of the highest concerns in the healthcare system, especially in medical rehabilitation settings. However, there is a lack of instruments for the assessment of risk falls in the context of musculoskeletal rehabilitation. Methods. This retrospective multisite study aimed to assess the sensitivity and specificity of four fall risk assessment tools (the Functional Independence Measure, the Fall Risk Assessment, the Schmid Fall Risk Assessment Tool, and the ePA-AC) in predicting falls in patients admitted to musculoskeletal rehabilitation in Swiss inpatient facilities. Results. The data relative to 6970 individuals (61.5% females) were analyzed and 685 (9.83% of patients) fall events were registered. The area under the curve (AUC) relative to the Functional Independence Measure was 0.689, 0.66 for the Fall Risk Assessment, 0.641 for the Schmid Fall Risk Assessment Tool, and 0.675 for the ePA-AC. Among the four tools, the Functional Independence Measure had an acceptable discriminatory power in distinguishing between significant events (i.e., patients’ falls) and non-events (no falls). Conclusion. None of the assessed tools showed highly satisfying levels of statistical sensitivity or sensibility. However, the Functional Independence Measure could be used to assess the fall risk assessment in musculoskeletal rehabilitation settings, although with some caution, since this questionnaire was not designed for this diagnostic purpose. We strongly suggest urgently designing a tool for risk assessment that is specific to this population and the rehabilitative setting.

## 1. Introduction

The World Health Organization (2021) defines a fall as “*an event which results in a person coming to rest inadvertently on the ground or floor or other lower level. Fall-related injuries may be fatal or non-fatal though most are non-fatal*” [1]. Morse [2] categorizes falls as (i) accidental, when patients fall unintentionally, so the fall risk cannot be identified before the event, even through standardized tools; (ii) unanticipated physiologic, which occurs when the fall is caused by physical conditions that cannot be predicted in advance, such as bone break or seizure; and, (iii) *anticipated physiologic*, which are clinical factors that increase the risk of fall, such as a prior fall, weak or impaired gait, use of a walking aid, intravenous access, or impaired mental status. Of course, patient falls represent one of the highest concerns in the healthcare system, especially in medical rehabilitation settings, where a higher incidence rate of falls is registered when compared with acute medical units (38% vs. 2%) [3]. Studies investigating falls among orthopedic inpatients (i.e., [4,5]) reported fall rates of 0.9–1% at the admissions prior to surgery. Institutions aim to design fall protective and preventive interventions (i.e., [2,5,6,7,8]): managing patients’ underlying fall risk may be the key to fall prevention [9]. Nevertheless, diagnostic tools, such as questionnaires or scales, are mandatory to establish *a priori* risk of falls, specifically at the beginning of hospitalization. However, in the literature, there is a lack of tools for the assessment of fall risk in rehabilitation settings since most of them have been developed and tested for acute settings, limiting their applicability in the rehabilitation context. The only exception is the Casa Colina Fall Risk Assessment Scala by Rosario and colleagues [9,10], which is specifically designed for stroke and brain injury. This lack is critical: intuitively, a higher risk of falling may be associated with poor walking ability, pain, fractures, the use of crutches and wheelchairs, and overall with all these conditions in which we may observe mobility limitations.

Designing specific and valid assessment tools is complex and requires time; moreover, it would be useful to have some evidence about what questions may be more suitable to be included in the assessment. Furthermore, in the meantime, clinicians may use other published clinical tools to collect information about the risk of falls in their patients; nevertheless, they should be informed about their level of efficacy in predicting such a risk. In this retrospective study, we analyzed the level of accuracy in predicting fall risks of some questionnaires used in musculoskeletal rehabilitation settings in Switzerland, according to local guidelines. Specifically, we analyzed the Functional Independence Measure (FIM) [11,12], which is one of the most widely used disability and dependence assessment instruments in rehabilitation medicine in comparison with the Fall Risk Assessment by the Ubiquity Quality Healthcare Group, Inc; The Schmid Fall Risk Assessment Tool [13]; and the AcuteCare ePA-AC [14]. Notably, all these scales were developed for acute clinical settings; moreover, these scales did not aim to measure specifically fall risk. We aimed to provide evidence about the levels of sensitivity and specificity of these scales in identifying musculoskeletal patients at risk of falling in a rehabilitation setting.

## 2. Methods

This study was approved by the Swiss Association of Research Ethics Committees (2023-01193; Rif. 4397) and it was conducted according to the Code of Ethics of the World Medical Association (Declaration of Helsinki). Data for this research were retrospectively extracted from electronic records furnished by the Swiss National Association for Quality Development in Hospitals and Clinics (ANQ), which coordinates and undertakes reviews of the quality of inpatient acute care, rehabilitation, and psychiatric treatment in Switzerland (https://www.anq.ch/en/ (accessed on 1 September 2023)). Extracted data referred to the time ranging from 1 January 2014 to 31 December 2017. The data referred to Clinica Hildebrand–Brissago and Clinica di Riabilitazione dell’Ente Ospedaliero Cantonale (Novaggio and Faido), which are part of ReHa Ticino (Faido, Switzerland). Data were relative to those patients with a musculoskeletal diagnosis. For each of them, we determined if they fell during rehabilitation treatment. For all included patients, we used the Cumulative Illness Rating Scale (CIRS) [15,16] to assess the presence and cumulative severity of pre-existing pathologies. The scale consists of 14 health-related domains. Each item is scored on a 5-point ordinal scale, ranging from a score of 0 (i.e., no impairment to that organ or system) to a score of 4 (i.e., extremely severe problem and/or immediate treatment required and/or organ failure and/or severe functional impairment). We computed the Severity Index–SI, as the mean of scores relative to the first 13 items, and the Comorbidity Index–CI, as the sum of the first 13 items in which participants reported a score equal to or over 2 (higher score of 13). The score relative to the psychiatric domain was independently reported. 

For all included patients, the scores relative to the following risk assessment tools, as part of a standard clinical assessment performed by physicians according to the ANQ guidelines, were extracted. 

*The Functional Independence Measure* (FIM) [11,12] consists of an 18-item, seven-level, ordinal scale. It documents the amount of help that a patient needs to perform 18 tasks, grouped into two subscales: motor and cognitive. The motor subscale includes eating, grooming, bathing, dressing the upper body, dressing the lower body, toileting, bladder management, bowel management, bed/chair/wheelchair transfers, toilet transfers, bath/shower transfers, walking/using a wheelchair, and using stairs. Each item is scored on a 7-point ordinal scale, ranging from a score of 1 (i.e., total assistance/not testable) to 7 (i.e., complete independence). The motor subscale is the sum of the individual motor subscale items and gives a value between 13 and 91. The cognition subscale includes comprehension, expression, social interaction, problem-solving, and memory. Again, each item is scored on a 7-point ordinal scale, ranging from a score of 1 (i.e., total assistance/not testable) to 7 (complete independence). The sum of the individual cognition subscale items results in the cognition subscale score, which gives a value between 5 and 35. Finally, a total FIM score, which is the sum of the two subscales scores, can be a value between 18 and 126. The higher the score, the higher the individual level of functional independence. 

The *Fall Risk Assessment* is provided by the Ubiquity Quality Healthcare Group, Inc. It was designed for the geriatric population in assisted living facility environments to identify those individuals at risk and implement interventions to reduce the frequency and severity of falls. It is composed of nine (scale from 0 to 4) items assessing the levels of consciousness/mental status, history of falls relative to the previous three months, ambulation/elimination status, vision status, gait and balance, orthostatic changes, medications, predisposing disease, and equipment. The sum of all items can vary from 0 (no risk of fall) to 25 (high risk of fall). 

The *Schmid Fall Risk Assessment Tool* [13] is used to categorize the risk of falling by assessing certain patient characteristics in five domains: mobility, mentation, elimination, prior fall history, and current medications. The tool scoring system ranges from 0 to 6, with 0 being no identified risk and scores of 3 or greater identifying a patient as at risk of falling. Thus, a higher score suggests a higher fall risk. 

The *ePA-AC* [14] is a 56-item nursing instrument developed to measure abilities and impairments in 11 health domains (e.g., state of consciousness or motor skills). Individual items are rated dichotomously or on 4-point Likert scales and can be used to calculate a variety of scores, including fall risk. Specifically, this risk is computed according to nine factors: walking; walking difficulties; balance difficulties; recent falls; fall event; voiding urgency; (time, spatial, personal) orientation; drugs increasing the fall risk; and visual abilities. For each domain, a score is assigned. A lower score suggests a higher fall risk. 

### Analyses

Data were summarized using the mean with standard deviation for continuous data and percentages for discrete variables. We computed the receiver operating characteristic (ROC) to assess the discriminatory capability of the four fall risk scales (i.e., the Functional Independence Measure, the Fall Risk Assessment, the Schmid Fall Risk Assessment Tool, and the ePA-AC). Specifically, we computed sensitivity, specificity, positive predictive value, negative predictive value, accuracy, and diagnostic odds ratio (95% confidence interval) [17]. Accuracy was scored with the following formula: (number of false positive cases + number of false negative cases)/total cases [17]. The area under the ROC curve (AUC) summarized the overall performance of the fall risk scales: it defined the probability that a classifier would rank a randomly chosen patient that fell higher than a randomly chosen patient that did not fall. Greater tool discrimination was reflected by sensitivity, specificity, predictive value, and AUC closer to 1. Hosmer and Lemeshow [18] provided the following classification system for the AUC: 0.7 ≤ AUC < 0.8 = ‘Acceptable discrimination’; 0.8 ≤ AUC < 0.9 = ‘Excellent discrimination’; AUC ≥ 0.9 = ‘Outstanding discrimination’. We compared the AUC of each fall risk scale with the ideal AUC of 0.5 (i.e., it ranks a random positive example higher than a random negative example 50% of the time). The AUC was also used to determine the cut-off score.

## 3. Results

### Participants

Data relative to 6970 patients (61.5% females) were extracted and 685 (9.83% of patients) fall events were registered. In Table 1, we reported details about the data distribution within the time range. In Table 2, we show the data distribution according to the different age classes. In Table 3, we report the percentage of assessed individuals for the different musculoskeletal diagnoses, according to the International Classification of Diseases [19].

In Table 4, for the four tools, we report the value relative to the AUCs and diagnostic odds ratio (95% confidence interval), while in Table 5, we report the statistical values of sensitivity, specificity, positive predictive value, negative predictive value, and accuracy. 

According to the results reported in Table 4, the AUC relative to the Functional Independence Measure [11,12] was 0.689 (95% CI from 0.659 to 0.719); 0.66 for the Fall Risk Assessment (Ubiquity Quality Healthcare Group, Inc.) (95% CI from 0.581 to 0.74), 0.641 for the Schmid Fall Risk Assessment Tool [13] (95% CI from 0.605 to 0.676), and 0.675 for the ePA-AC [14] (95% CI from 0.591 to 0.759). Moreover, all the AUCs were significantly different from AUC = 0.5 (*p* always <0.001). As shown in Figure 1, considering the 95% CI upper limits reported in Table 4, the Functional Independence Measure [11,12], the Fall Risk Assessment (Ubiquity Quality Healthcare Group, Inc.), and the ePA-AC [14]), but not the Schmid Fall Risk Assessment Tool [13], had an acceptable discriminatory power in distinguishing between significant events (i.e., patient fall) and non-events (no fall).

## 4. Discussion

In this article, we analyzed the statistical properties of four scales (the Functional Independence Measure [11.12]; the Fall Risk Assessment by the Ubiquity Quality Healthcare Group, Inc; the Schmid Fall Risk Assessment Tool [13]; and the ePA-AC [14]) in predicting the fall risk of patients with a musculoskeletal diagnosis at the beginning of a rehabilitative procedure. Notably, none of these scales was designed for assessing musculoskeletal patients and rehabilitation settings. However, two of them (i.e., the Fall Risk Assessment by the Ubiquity Quality Healthcare Group, Inc and the Schmid Fall Risk Assessment Tool [13]) were specifically designed for the risk assessment. The Functional Independence Measure [11,12] is one of the most widely used disability and dependence assessment instruments in rehabilitation medicine, and there are several studies (for a review, see [20]) adopting this questionnaire to identify patients at risk of falling. Finally, we also included the outcome-oriented nursing assessment instrument of AcuteCare (ePA-AC) [14]: most of the time, nursing staff perform the clinical assessment of patients admitted to rehabilitative units. Thus, this tool was designed as a screening instrument to identify patient abilities or disabilities in acute inpatient settings to quantify relevant aspects of the need for nursing care, hospital management, and quality management [21].

According to our results, none of the assessed tools had highly satisfying levels of statistical sensitivity and sensibility. Thus, clinicians should take caution in adopting them in assessing the risk of falls in musculoskeletal rehabilitative settings. It may be observed that the score of the Functional Independence Measure [11,12] had an acceptable statistical power in distinguishing between patients who fell from patients who did not fall during recovery. This result is in agreement with that of some previous studies relative to mixed clinical diagnosis: lower scores on the scale predict an increased likelihood of fall [21,22,23,24,25,26,27]. Nevertheless, the scale was not able to predict the risk of falls in our musculoskeletal sample, mirroring what was observed in other contexts, such as geriatric [20] and stroke wards [19,20]. Thus, even if the Functional Independence Measure [11,12] is one of the most widely used disability and dependence assessment instruments in rehabilitation medicine, it may be not efficient in risk assessment in rehabilitation settings, and should be used with this aim and in this context with some caution. 

According to our results, the assessed tools showed no differences in terms of sensitivity. It may be observed that the Schmid Fall Risk Assessment Tool [13] showed the highest score (78.7%), followed by the Functional Independence Measure [11,12] (67.95%), then by the Fall Risk Assessment by the Ubiquity Quality Healthcare Group (59.09%), and finally the ePA-AC [14] (56.41%). However, the Schmid Fall Risk Assessment Tool [13] reported the lowest level of specificity in comparison with the other tools (46.91%). No differences emerged between the other three (ePA-AC [14] with 67.31%, FIM [11,12] with 62.94%, and the Fall Risk Assessment with 62.32%): they showed comparable levels of specificity. Crucially, the Schmid Fall Risk Assessment Tool [13] had the lowest level of accuracy (50.32%) in comparison with other tests: accuracy measures the degree of veracity of a diagnostic test on a condition, in this case, *the fall* in our sample of individuals with a musculoskeletal diagnosis during rehabilitation treatment. The other three scales were not significantly different (ePA-AC [14] with 66.61%, FIM [11,12] with 63.38%, and the Fall Risk Assessment with 62.02%) in their level of accuracy. 

In conclusion, even though some risk assessment tools are available in the literature [28], not one is specific to musculoskeletal rehabilitation settings. In rehabilitation facilities, clinicians primarily aim to increase an individual’s physical functions and mobility, even if patients suffer from cognitive difficulties, to facilitate a safe discharge. This goal may increase the risk of falls during recovery in a rehabilitation setting in comparison with acute ones. Moreover, individuals with a musculoskeletal impairment may face difficulties and have rehabilitative goals that would be extremely different from those of patients with other clinical diseases. For these two reasons, we strongly recommend designing a risk assessment tool specific to this population and a rehabilitative setting. An example is the Casa Colina Fall Risk Assessment Scala [9,10] in the context of rehabilitation for stroke and brain injury. 

Finally, we underline some limitations in our study. We verified the data relative to two inpatient rehabilitation facilities. Nevertheless, because both are part of the ReHa Ticino (Switzerland), they shared rehabilitation assessments and protocols. We collected data from a very large sample; however, because of the retrospective nature of our study, we had a selected number of descriptive and clinical factors to describe our population. Moreover, all the data collected referred to timing before the beginning of the COVID-19 pandemic, because of which the level of efficacy of the diagnostic tools may have changed [9], requiring updated data. Finally, in this article, we used receiver operating characteristic (ROC) curve analysis to analyze the effectiveness of the different tools in assessing the risk of fall; according to this analysis, a cut-off value can be selected (as reported in Table 5 for the assessed tools) [17]. However, the Bayesian approach which provides information about how a test result would change the odds (and thus probability) of a disease could be used in further research [29].

## Figures and Tables

**Figure 1 healthcare-11-02805-f001:**
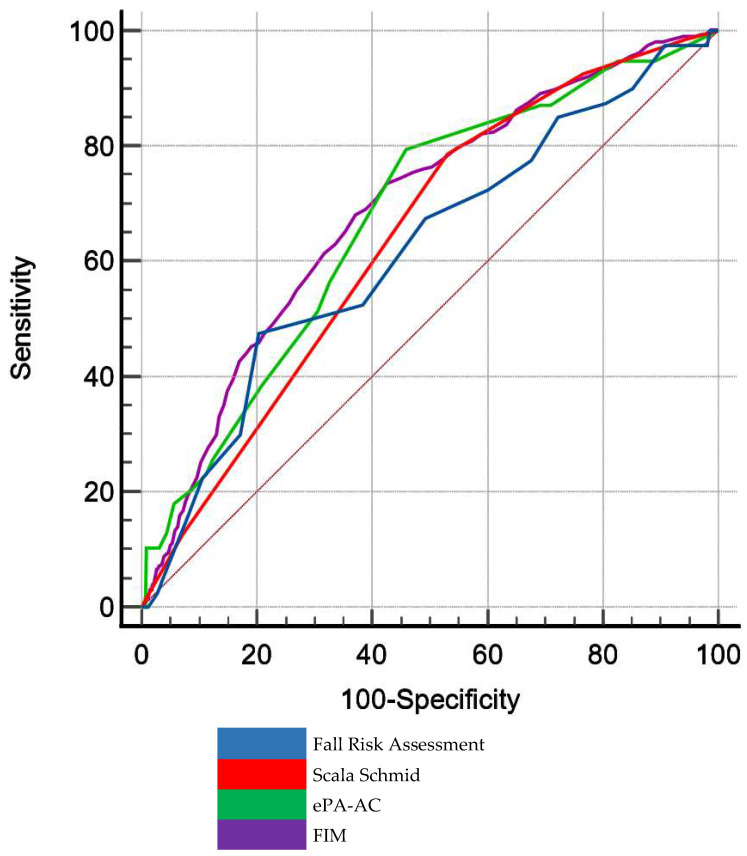
The receiver operating characteristic curve for the Fall Risk Assessment (in blue), the Schmid Fall Risk Assessment (in red), the ePA-AC (in green), and the Functional Independence Measure (in purple). On the x-axis, the level of specificity (i.e., the false positive rate); on the y-axis, the level of sensitivity (i.e., the true positive rate).

**Table 1 healthcare-11-02805-t001:** Data distribution over the time range. For each year and for the total sample, we report the number of assessed patients and the number and percentage of patients who fell.

	2014			2015			2016			2017			Overall		
	Total	Fallen	% Fallen	Total	Fallen	% Fallen	Total	Fallen	% Fallen	Total	Fallen	% Fallen	Total	Fallen	% Fallen
N	1533	161	10.5	1588	181	11.4	1821	193	10.6	2028	150	7.4	**6970**	**685**	**9.83**

**Table 2 healthcare-11-02805-t002:** Data distribution (N = sample size and relative percentage), mean (M), and standard deviation (SD) relative to the enrolled individuals across the different age classes (in years) over the time range and for the entire sample.

	2014				2015				2016				2017				Total			
Age Class	N	%	M	SD	N	%	M	SD	N	%	M	SD	N	%	M	SD	N	%	M	SD
(in Years)																				
≤49	124	8.09	42.14	7.12	119	7.49	41.17	7.07	131	7.19	41.24	7.52	171	8.43	40.08	8.5	545	7.82	41.06	7.68
50–59	176	11.48	55.06	2.83	159	10.01	54.99	2.93	182	9.99	55.03	2.94	207	10.21	55.03	2.87	724	10.39	55.03	2.89
60–69	247	16.11	65.48	2.88	279	17.57	65.29	2.77	330	18.12	65.56	2.77	344	16.96	65.33	2.75	1200	17.22	65.42	2.79
70–79	532	34.7	74.67	2.86	531	33.44	74.64	2.79	612	33.61	74.82	2.71	699	34.47	74.84	2.81	2374	34.06	74.75	2.79
80–89	406	26.48	83.46	2.54	446	28.09	83.52	2.59	501	27.51	83.7	2.76	535	26.38	83.66	2.65	1888	27.09	83.6	2.64
≥90	48	3.13	91.79	1.97	54	3.4	92.61	2.34	65	3.57	92.03	2.16	72	3.55	92.78	2.8	239	3.43	92.34	2.39
*Total*	*1533*	*100*	*71.17*	*13.1*	*1588*	*100*	*71.63*	*13.14*	*1821*	*100*	*71.81*	*13.04*	*2028*	*100*	*71.24*	*13.76*	*6970*	*100*	*71.46*	*13.29*

**Table 3 healthcare-11-02805-t003:** Relative percentage of assessed individuals for musculoskeletal diagnosis according to the International Classification of Diseases (WHO, 2007).

Diagnosis	Percentage
Injury (selected sections S and T)	22.62
Dorsopathies (M40–M54)	15.16
Gonarthrosis (M17)	13.87
Other disorders of the musculoskeletal system and connective tissue (M95–M99, R26, R52)	9.67
Coxarthrosis (M16)	9.52
Other diseases	7.99
Neoplasms of central nervous system (selected sections C and D)	4.86
Soft tissue disorders (M60–M79)	4.12
Osteopathies and chondropathies (M80–M94)	2.77
Complications of internal orthopedic prosthetic devices, implants, and grafts (T84) and complications peculiar to reattachment and amputation (T87)	2.73
Arthropathies (M00–M25)	2.40
Spastic quadriplegic cerebral palsy and other paralytic syndromes (G80–G83)	1.49
Degenerative diseases of the nervous system (G10–G32)	0.59
Nerve, nerve root, and plexus disorders (G50–G59)	0.36
Diseases of myoneural junction and muscle (G70–G73)	0.30
Cerebrovascular diseases (I60–I69)	0.22
Demyelinating diseases of the central nervous system (G35–G37)	0.16
Systemic connective tissue disorders (M30–36)	0.09
Polyneuropathies and other disorders of the peripheral nervous system (G60–G64)	0.07
Other disorders of the nervous system (G90–G99)	0.06
Episodic and paroxysmal disorders (G40–G47)	0.06
Inflammatory diseases of the central nervous system (G00–G09)	0.04
Not specified	0.87

**Table 4 healthcare-11-02805-t004:** For the assessed tools, we report the AUCs and diagnostic odds ratio (95% confidence interval.

	Area under the Curve (AUC)	Standard Error	*p* Value	95% CI	
				Lower Limit	Upper Limit
Fall Risk Assessment	0.66	0.041	<0.001	0.581	0.74
Schmid	0.641	0.018	<0.001	0.605	0.676
ePA-AC	0.675	0.043	<0.001	0.591	0.759
Functional Independence Measure	0.689	0.015	<0.001	0.659	0.719

**Table 5 healthcare-11-02805-t005:** Diagnostic parameters for the assessed tools.

Fall Risk Assessment				Schmid Fall Risk Assessment Tool				ePA-AC				Functional Independence Measure			
Cut-off	Not fallen	Fallen	Total	Cut-off	Not fallen	Fallen	Total	Cut-off	Not fallen	Fallen	Total	Cut-off	Not fallen	Fallen	Total
<13	263	18	281	< 3	842	46	888	>27	381	17	398	>88	2060	100	2160
≥13	159	26	185	≥ 3	953	170	1123	≤27	185	22	207	≤88	1213	212	1425
Total	422	44	466	Total	1795	216	2011	Total	566	39	605	Total	3273	312	3585
	**Value**	**95% CI**			**Value**	**95% CI**			**Value**	**95% CI**			**Value**	**95% CI**	
**Sensitivity (%)**	59.09	43.25-73.66		**Sensitivity (%)**	78.7	72.64–83.97		**Sensitivity (%)**	56.41	39.62–72.19		**Sensitivity (%)**	67.95	62.46–73.09	
**Specificity (%)**	63.32	57.51-66.96		**Specificity (%)**	46.91	44.58–49.25		**Specificity (%)**	67.31	63.28–71.17		**Specificity (%)**	62.94	61.26–64.6	

## Data Availability

The data presented in this study are available on request from the corresponding author. The data are not publicly available due to ethical restriction.

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
