# Peer review of "Predicting Falls in Musculoskeletal Rehabilitation: A Retrospective Multisite Study"

_healthcare, 2023, doi:10.3390/healthcare11202805_

Round 1

Reviewer 1 Report

The authors performed a retrospective analysis of a large dataset comparing several fall risk assessment tools. The sensitivity and specificity of these tools were quite low.
The manuscript is technically sound, but the results were disappointing.

Due to the low sensitivity and specificity values, none of these instruments provide a clinically relevant value. In my opinion, this is not emphasized enough in the discussion and in the abstract.

Major comments:
The abstract and the discussion have to discuss the sensitivity and specificity values regarding their clinical value and potential use.
Also the title (Predicting falls in musculoskeletal rehabilitation: a retrospective multisite study) should give the main information, i.e. that falls were not relevantly predicted
The statistical analysis should evaluate different cut off values for each test. Perhaps the value for a clinical use can be improved by applying other cut off values.

There are some minor comments:

L 13 I suggest omitting "patient" at the beginning of the sentence.
L21 The "acceptable discrimination power" should be reported here in more detail. The other tools should also be reported here.
Table 1: typo: 2017 instead of 201
Table 2: mean and SD and range of age of each year should be reported
L 174, I think it should be AUC=0.5
Fig 1, all four curves should be shown in one figure for better comparison.

Author Response

Manuscript ID: healthcare-2644953

ARTICLE: Predicting Falls in Musculoskeletal Rehabilitation: A Retrospective Multisite Study.

The authors performed a retrospective analysis of a large dataset comparing several fall risk assessment tools. The sensitivity and specificity of these tools were quite low.
The manuscript is technically sound, but the results were disappointing.

Due to the low sensitivity and specificity values, none of these instruments provide a clinically relevant value. In my opinion, this is not emphasized enough in the discussion and in the abstract.

REPLY: We thank the Reviewer for this very important comment. Both in the abstract (lines 20-21) as well as in the Discussion (lines 205-206), we underlined that none of the assessed tool report high satisfying level of statistical sensitivity and sensibility.

Major comments:
The abstract and the discussion have to discuss the sensitivity and specificity values regarding their clinical value and potential use.
Also the title (Predicting falls in musculoskeletal rehabilitation: a retrospective multisite study) should give the main information, i.e. that falls were not relevantly predicted
The statistical analysis should evaluate different cut off values for each test. Perhaps the value for a clinical use can be improved by applying other cut off values.

REPLY: As previously stated, in the new version of the manuscript, in the abstract (lines 20-21) as well as in the Discussion (lines 205-206), we underlined that none of the assessed tool report high satisfying level of statistical sensitivity and sensibility. Instead, we prefer to mantain the current title, to avoid a long sentence. For each test, we reported the cut-off in Table 4. This cut-off result from the ROC analyses. About the Reviewer’s proposal to use test the statistical properties of other cut-offs, we are not sure on which ground we would choice them; indeed, unfortunately, in the previous literature there is no reference about this topic especially in the clinical population presented in this paper. Neverthless, considering that we applied the standard approach as well as our results (none of the assessed tools reports satisfying level of statistical sensitivity and sensibility), this analysis can be confounding, while we would encourage to create ad-hoc assessment for fall risk in muscholoscheletrical settings. On the other hand, in the future different statistical approachs can be used such as the Bayesian approach. We report this proposal in the final part of our Discussion (lines 245-251).

 There are some minor comments:

L 13 I suggest omitting "patient" at the beginning of the sentence.

REPLY: Done. Thank you.

L21 The "acceptable discrimination power" should be reported here in more detail. The other tools should also be reported here.

REPLY: According to this comment, for all the tools we reported the details about the AUC.

Table 1: typo: 2017 instead of 201

REPLY: Thank you for having noticed this typo. We changed the text.

Table 2: mean and SD and range of age of each year should be reported

REPLY: We thank the Reviewer for this comment; we included the required information in Table 2.

L 174, I think it should be AUC=0.5

REPLY: Thank you for having noticed this typo. We changed the text.

Fig 1, all four curves should be shown in one figure for better comparison.

REPLY: Following this suggestion, we provided a new version of Figure 1, in which all four curves were plotted. We thank the Reviewer for this suggestion.

Reviewer 2 Report

Congratulations for the work done, I am enclosing some minor corrections to your manuscript:

INTRODUCTION: The first sentence of the WHO is not referenced, please do so even though it is a web page. On the other hand, the number of references is low and of low quality. In this subject there are systematic reviews with meta-analysis with much scientific rigor. I recommend that you perform a more advanced search on this construct and add them.

MATERIALS AND METHODS: Both the description and the registration and analysis of data are correct for this scientific research.

RESULTS: The graphs are very helpful to the reader in interpreting the data.

DISCUSSION: As in the introduction, more references are needed to make a better comparison and discussion of the data. As for the limitations described, they are consistent with the type of the manuscript.

Once again, I hope that with these slight modifications your manuscript can be published. 

Author Response

Congratulations for the work done, I am enclosing some minor corrections to your manuscript:

INTRODUCTION: The first sentence of the WHO is not referenced, please do so even though it is a web page. On the other hand, the number of references is low and of low quality. In this subject there are systematic reviews with meta-analysis with much scientific rigor. I recommend that you perform a more advanced search on this construct and add them.

MATERIALS AND METHODS: Both the description and the registration and analysis of data are correct for this scientific research.

RESULTS: The graphs are very helpful to the reader in interpreting the data.

DISCUSSION: As in the introduction, more references are needed to make a better comparison and discussion of the data. As for the limitations described, they are consistent with the type of the manuscript.

Once again, I hope that with these slight modifications your manuscript can be published. 

REPLY: We are grateful to the Reviewer for the time devoted in reviewing our manuscript. As requested, we enlarged our reference list, including some recent reviews on the topic. Moreover, we edited the paper considering the comments received from the other Reviewers. We believe that our manuscript has improved.

Reviewer 3 Report

The authors  analyzed the statistical properties of four scales/tools in predicting fall risk in patients with a musculoskeletal impairment at the beginning of the rehabilitative procedure.

Overall it is a well written paper proving evidence regarding the usefulness of those tools.

Some comments follow

Methods section

It is not clear if you used retrospective data only from the clinics you describe or you contacted individual patients to utilise your tools. Please clarify. 

Why have you chosen those tools? please give a brief statement on each one for their usefuleness and statistical merit as some were not designed as risk assessment tools. Perhaps you might also consider inserting a table including all tools with result's range and brief meaning. That might be preferable to the way you have presented them in your paper

Results section

Table 1: the number in the total column is misleading the reader as even though it is a 4 figure number, it reads as a 3 figure and a separate number below. consider revising (smaller font?)

Discussion section

a clear conclusion and clinical significance of fidnings appear to be missing from this section. 

Author Response

Manuscript ID: healthcare-2644953

ARTICLE: Predicting Falls in Musculoskeletal Rehabilitation: A Retrospective Multisite Study.

The authors  analyzed the statistical properties of four scales/tools in predicting fall risk in patients with a musculoskeletal impairment at the beginning of the rehabilitative procedure.

Overall it is a well written paper proving evidence regarding the usefulness of those tools.

Some comments follow

REPLY: We thank the Reviewer for the time devoted in reviewing our manuscript.

Methods section

It is not clear if you used retrospective data only from the clinics you describe or you contacted individual patients to utilise your tools. Please clarify.

REPLY. We thank the Reviewer for this comment. In the method we clarified that the data were extracted from the eletronical records furnished by the Swiss National Association for Quality Development in Hospitals and Clinics (ANQ) about the involved clinical centers. Thus, no patient was contacted for the data.

 Why have you chosen those tools? please give a brief statement on each one for their usefuleness and statistical merit as some were not designed as risk assessment tools. Perhaps you might also consider inserting a table including all tools with result's range and brief meaning. That might be preferable to the way you have presented them in your paper

REPLY. As reported in the methods, the risk assessment tools were part of a standard clinical assessment performed by physicians in the involved hospitals according to the ANQ guidelines.

Results section

Table 1: the number in the total column is misleading the reader as even though it is a 4 figure number, it reads as a 3 figure and a separate number below. consider revising (smaller font?)

REPLY. We are sorry, but we were not able to understand this comment. Neverthless, we revised Tables and Figure 1 to be more readable. We hope that these actions solved the Reviewer’s comment.

Discussion section

a clear conclusion and clinical significance of fidnings appear to be missing from this section. 

REPLY. Because of this valuable comment, we included a new sentence in our Discussion, in which we specified that none of the assessed tools had highly satisfying levels of statistical sensitivity and sensibility. Indeed, it is the main take-home message from our study.

Round 2

Reviewer 1 Report

I thank the authors for addressing my concerns. I suggest to accept the manuscirpt in its current form.